# Ocular biometric parameters changes and choroidal vascular abnormalities in patients with neurofibromatosis type 1 evaluated by OCT-A

**Aldo Vagge[1,2], Paolo Corazza[1], Lorenzo Ferro Desideri[1]\*, Paola Camicione[3], Giulia Agosto[4], Roberta Vagge[4], Calevo Maria Grazia[5], Adriano Carnevali[6], Giuseppe Giannaccare[6], Massimo Nicolò[1,2], Carlo Enrico Traverso[1,2]**

1 Department of Neuroscience, Rehabilitation, Ophthalmology, Genetics, Maternal and Child Health (DiNOGMI), Eye Clinic of Genoa, Policlinico San Martino, University of Genova, Genova, Italy, 2 IRCCS Ospedale Policlinico San Martino, Genova, Italy, 3 Department of Ophthalmology, Giannina Gaslini Institute, Genoa, Italy, 4 Department of Neurosciences, Rehabilitation, Ophthalmology, Genetics, Maternal and Child Health (DiNOGMI), School of Orthoptists and Ophthalmology Assistants, University of Genoa, Genoa, Italy, 5 Epidemiology and Biostatistics Unit, Giannina Gaslini Institute, Genoa, Italy, 6 Department of Ophthalmology, University of "Magna Græcia", Catanzaro, Italy

\* lorenzoferrodes@gmail.com

## Abstract

### Purpose

To analyze ocular biometric parameters alterations of the posterior pole and choroidal abnormalities in patients with neurofibromatosis type 1 (NF1) by adopting multimodal imaging, especially focusing on the role of novel diagnostic devices like swept-source optical coherence tomography angiography (SS-OCTA).

### Methods

In this prospective, case-controlled study, patients with NF1 and age-matched control subjects were quantitatively analyzed by using multimodal imaging. All the subjects underwent confocal scanning laser ophthalmoscopy (SLO), SS-OCT and SS-OCTA examinations.

### Results

SS-OCT analysis revealed a lower macular retinal nerve fiber layer (RNFL) thickness in patients with NF1 compared with those with suspected NF1 ($95.0\pm15.9$ vs $109.7\pm11.3$ µm; $P = 0.001$) and control subjects ($106.8\pm14.4$ µm, $P = 0.003$). Retinal thickness was significantly lower in NF1 patients compared to those with suspected NF1 ($280.7\pm23.0$ vs $304.2\pm15.3$ µm; $P < 0.001$) and control subjects ($298.7\pm23.8$ µm, $P = 0.003$). The mean vascular flow area of the SCP was significantly higher in patients with NF1 ($42.6\pm2.2\%$) and suspected NF1 ($43.1\pm2.5\%$) compared to control subjects ($41.0\pm2.0\%$; respectively, $P = 0.017$ and $P = 0.002$). In the second choroidal layer, the flow area was significantly lower in patients with NF1 compared to control subjects ($45.4\pm4.8$ vs $49.0\pm4.0\%$,; $P = 0.011$).

**Data Availability Statement:** All relevant data are within the paper. Additional data cannot be shared publicly because of privacy policy adopting by

Clinica Oculistica. These data are available upon request from Dr. Maria Musolino (Senor Research Assistant, maria.musolino@unige.it), who can field any data inquiries from fellow researchers. Dr. Musolino is responsible for data management and sharing at the University Eye Clinic of Genoa (DINOGMI), Italy.

**Funding:** No - The funders had no role in study design, data collection and analysis, decision to publish, or preparation of the manuscript.

**Competing interests:** No - The authors have declared that no competing interests exist.

## Conclusions

Retinal thicknesses alterations and choroidal nodules are described as ocular manifestations in patients with NF1. In addition, OCTA could represent an important novel advanced imaging technique, capable of detecting early altered retinal and choroidal vascular flow area in patients with NF1.

## Introduction

Neurofibromatosis type 1 (NF1), also known as von Recklinghausen's disease, is one of the most frequent autosomal dominant disorders, showing an incidence of 1 out of 2500–3000 births [1]. This genetic neurocutaneous disorder is due to the mutation of the NF1 gene, displaying a tumor suppressor role and situated on chromosome 17q11.2. NF1 encodes for neurofibromin, a protein involved in the RAS signal transduction pathway, whose loss ultimately leads to constitutive downstream signaling and abnormal cell growth [2, 3].

Nowadays, the criteria for the diagnosis of NF1 still remain clinical; they were first defined in 1988 by the National Institutes of Health (NIH) and comprehend the contemporary presence of these following criteria: six or more café-au-lait spots, two or more cutaneous neurofibromas, one or more plexiform neurofibromas, axillary or inguinal freckling, optic glioma, two or more iris Lisch nodules, characteristic skeletal dysplasia and a first-degree relative with NF1 diagnosis; however, in some centers genetic assessment is commonly performed in order to confirm NF1 diagnosis [4, 5].

Although Lisch nodules represent the most common ocular manifestation of the disease, several other ocular conditions such as optic gliomas, eyelid neurofibromas and café-au-lait spots, choroidal nevi, choroidal nodules and congenital glaucoma have been described in association with NF1 [6]. In this regard, it has been shown by near-infrared reflectance (NIR) imaging that choroidal abnormalities may occur in 100% of the patients with NF1 [7]. For this reason, choroidal nodules have been investigated as new possible diagnostic criteria of NF1 [8]. The presence of choroidal nodules, detected as bright patchy areas by NIR, has previously been reported in 70% of pediatric patients with NF1 [3]. Mostly undetectable by performing normal ophthalmological examination and fluorescein angiography (FA), choroidal nodules may also be observed with indocyanine green angiography (ICGA); however, because of the invasive nature of this latter advanced imaging technique, this should not be considered as a first line strategy like NIR in detecting choroidal abnormalities [9].

Furthermore, in patients with NF1 it has been reported the association between choroidal nodules and retinal vascular abnormalities (RVAs) [10]. RVAs are described as small tortuous venules with a "corkscrew" appearance and their reported prevalence measured by FA or EDI-OCT ranges between 35% and 37.5% in patients with NF1 [11, 12]. More recently, the introduction of novel, non-invasive devices like OCT angiography (OCTA) has enabled us to better characterize retinal and choroidal vasculature [13]. Parrozzani *et al.* detected RVA in only 6.1% of the patients with NF1 by OCTA.

The aim of our study is to evaluate the ocular biometric parameters changes of the posterior pole and the choroidal vascular abnormalities in patients affected with NF1. Furthermore, we will discuss the possible additional role of new devices like OCTA in the early diagnosis of NF1 by comparing the retinal and choroidal vascular flow area between the examined group and healthy subjects of the same age.

## Patients and methods

This prospective, cross-sectional study was performed at the University Eye Clinic of Genoa, Department of Neuroscience, Rehabilitation, Ophthalmology, Genetics, Maternal and Child Health (DINOGMI), IRCCS Ospedale Policlinico San Martino in collaboration with IRCCS Istituto Giannina Gaslini, Genoa, Italy.

Informed consent was obtained verbally from all participants and/or their families (By the parents in case of minors) and this study was done in accordance with the Declaration of Helsinki and was approved by the local Institutional Ethics Committee (Ethics Committee of DINOGMI, IRCCS Ospedale Policlinico San Martino).

All the subjects presenting a stringent diagnosis of NF1 in accordance with the above-mentioned NIH criteria were included in this study; by contrast, the control group was represented by age-matched healthy subjects. In addition, age-matched subjects presenting less than 2 NIH diagnostic criteria of NF1 were enrolled in the study as "suspected NF1 group".

All the subjects whose posterior segment was not explorable by ophthalmological examination due to media opacity, refractive defects greater than ± 5 D (spherical equivalent) and any other ocular diseases including glaucoma and/or with borderline values of intraocular pressure (IOP) were ruled out from this prospective study. Moreover, also subjects presenting a medical history of ocular disease potentially involving retinal and choroidal tissues (such as uveitis, maculopathies, and other ocular congenital diseases) were excluded from the study. Diabetic patients were also ruled out from the study.

All the subjects underwent a comprehensive ophthalmological examination: firstly, best corrected visual acuity (BCVA) was measured, then a slit lamp examination in order to record the possible presence of Lisch nodules and a mydriatic indirect fundus biomicroscopy (with a 90-dioptre (D) lens in compliant patients or indirect fundus ophthalmoscopy with a 28-dioptre (D) lens in non-compliant patients) were performed. IOP measurement with Goldmann applanation tonometry was performed to all the patients.

All the participants underwent also confocal scanning laser ophthalmoscopy (SLO) with a Spectralis HRA+OCT device (Heidelberg Engineering, Heidelberg, Germany). SLO was performed in order to spot and count the number of choroidal lesions by setting NIR at 815 nm.

Then, all the subjects underwent swept-source OCT (SS-OCT) and OCTA (SS-OCTA) examinations, after having dilatated pupil with mydriatic tropicamide 1% ophthalmic drops. The SS-OCT device (DRI OCT Triton; Topcon Corporation) adopted in this study presents a central wavelength of 1050 nm, with an acquisition speed of 100,000 A-scans per second and an axial and transversal resolution of 7 and 20 μm in tissue, respectively. Each subject was examined with a 12-radial line (12 mm) acquisition centered on the fovea and three-dimensional (3D WIDE) acquisition protocol centered on the optic disc. Three-dimensional $4.5 \times 4.5$ mm$^2$ volumetric scans were obtained from 320 (horizontal) $\times$ 320 (vertical) A-scans. Choroidal abnormalities detected by SLO images were subsequently compared with SS-OCT scans in order to assess which was the degree of concordance between these two diagnostic tools. Furthermore, SS-OCTA scans were extrapolated from $4.5 \times 4.5$ mm$^2$ cubes with each cube including 320 clusters of four repeated B-scans centered on the fovea. Images from the retinal superficial capillary plexus (SCP), deep capillary plexus (DCP), choriocapillaris (CC) and choroid were considered for the analysis.

More in detail, the software automatically generated the OCTA images of the SCP, DCP, and CC. OCTA images of three deeper choroidal levels were taken by lowering the segmentation lines at the level of Sattler and Haller layers whose average depth has been referred to be approximately 259.70 and 125 μm, respectively, in healthy subjects. 3 different levels (L1, L2,

and L3) of the deep choroid were by setting the segmentation lines between 59.6 and 80.6 μm, 80.6 and 119.6 μm, and 119.6 and 137.8 μm from the Bruch's membrane, respectively [14].

Two investigators independently reviewed all scans to ensure correct segmentation and image quality for the post hoc assessment. Poor quality OCT-A scans with a low quality index than 30 were also excluded.

The number of choroidal nodules was manually calculated by expert ophthalmologists by analyzing NIR, OCTA and SS-OCT images. RVAs (retinal vascular abnormalities) consisted of small size second or third order tortuous venules, which were named 'corkscrew' retinal vessels. These abnormal vessels usually have a size of one or two disc diameters and end in a small tuff or disappear on the retinal surface [11].

The SPSS statistical software (SPSS Inc, Chicago, Illinois, USA) was used for data analysis. If both eyes were eligible, values from right eyes were used for statistical analysis. Values are expressed as mean ± standard deviation (SD). The $\chi^2$ test was used to compare dichotomous variables among patients with NF1, suspected NF1 and control subjects. A one-way ANOVA was used to compare continuous variables among the 3 groups. Post hoc comparisons between one group and another were performed using the Tukey's test A $P$ value $< 0.05$ was considered statistically significant.

## Results

Thirty-one patients with NF1 were enrolled and compared with 34-age matched healthy subjects and 25 subjects with suspected diagnosis of NF1. Clinical and demographical characteristics of enrolled patients are reported in Table 1. Patients with NF1 showed no significant difference in age compared to control subjects (22.7 ± 13.7 vs 24.9 ± 12.0 years, $P = 0.725$). Conversely, patients with suspected NF1 had a mean age of 13.7 ± 6.5 years and were significantly younger than patients with NF1 ($P = 0.012$) and control subjects ($P = 0.001$). No significant difference in sex distribution was observed among the 3 groups ($P = 0.034$).

In the NF1 group, 20 patients (64.5%) had Lisch nodules, compared to 6 patients (24.0%) in the suspected NF1 group and no patients in the control group ($P = 0.001$). Choroidal nodules were observed in 24 patients with NF1 (77.4%), 2 patients with suspected NF1 (8.0%) and no patients in the control group ($P < 0.001$). In patients with NF1, the mean number of choroidal nodules was 3.1 ± 2.9, and 8 patients (25.8%) had 5 or more nodules. Three patients with NF1 (9.7%) presented optic glioma, while no patients with suspected NF1 and no control subjects did. However, this difference was not significant ($P = 0.052$).

SS-OCT analysis revealed that patients with NF1 had a significantly lower macular retinal nerve fiber layer (RNFL) thickness compared to patients with suspected NF1 (95.0 ± 15.9 vs 109.7 ± 11.3 μm; $P = 0.001$) and control subjects (106.8 ± 14.4 μm, $P = 0.003$). Conversely, no significant difference in macular RNFL between patients with suspected NF1 and control subjects was observed ($P = 0.713$). Similarly, retinal thickness was significantly lower in NF1

**Table 1. Demographic and clinical characteristics of the enrolled patients.**

|  | NF1 (n = 31) | Suspected NF1 (n = 25) | Controls (n = 34) | *P* |
|---|---|---|---|---|
| Age (years) | 22.7 ± 13.7 | 13.7 ± 6.5 | 24.9 ± 12.0 | 0.001 |
| Sex (m/f) | 14/17 | 8/17 | 17/17 | 0.374 |
| Lisch nodules | 20 (64.5%) | 6 (24.0%) | 0 (0.0%) | <0.001 |
| Optic glioma | 3 (9.7%) | 0 (0.0%) | 0 (0.0%) | 0.052 |
| Choroidal nodules | 24 (77.4%) | 2 (8.0%) | 0 (0.0%) | <0.001 |

NF1: Neurofibromatosis type 1; LE: Left eye; RE: Right eye.

**Table 2. Retinal and choroidal thicknesses measured by SS-OCT in NF1 patients, suspected NF1 patients and healthy subjects.**

|  | NF1 (n = 31) | Suspected NF1 (n = 25) | Controls (n = 34) | $P^{*}$ |
|---|---|---|---|---|
| RNFL (μm) | 95.0 ± 15.9 | 109.7 ± 11.3 | 106.8 ± 14.4 | <0.001 |
| RT (μm) | 280.7 ± 23.0 | 304.2 ± 15.3 | 298.7 ± 23.8 | <0.001 |
| GCL (μm) | 42.9 ± 4.6 | 44.9 ± 4.1 | 43.8 ± 3.7 | 0.203 |
| CT (μm) | 147.0 ± 49.8 | 142.0 ± 43.3 | 151.7 ± 47.5 | 0.737 |

NF1: Neurofibromatosis type 1; RNFL: Retinal nerve fiber layer; RT: Retinal thickness; GCL: Ganglion cell layer; CT: Choroidal thickness.

[*] One-way ANOVA.

patients compared to those with suspected NF1 (280.7 ± 23.0 vs 304.2 ± 15.3 μm; $P < 0.001$) and control subjects (298.7 ± 23.8 μm, $P = 0.003$). Conversely, retinal thickness did not differ between patients with suspected NF1 and control subjects ($P = 0.598$). No significant differences of ganglion cell layer (GCL) thickness and choroidal thickness were observed among the 3 groups (both $P > 0.05$) (Table 2).

The results of the OCTA analysis in the 3 groups are reported in Table 3. The mean vascular flow area of the SCP was significantly higher in patients with NF1 (42.6 ± 2.2%) and suspected NF1 (43.1 ± 2.5%) compared to control subjects (41.0 ± 2.0%; respectively, $P = 0.017$ and $P = 0.002$). Conversely, no difference in the flow area of the SCP was observed between patients with NF1 and suspected NF1 ($P = 0.704$). No significant differences in the flow area of the DCP, CC, and first choroidal layer were observed between the 3 groups (always, $P > 0.05$). In the second choroidal layer, the flow area was significantly lower in patients with NF1 compared to control subjects (45.4 ± 4.8 vs 49.0 ± 4.0%; $P = 0.011$); while no differences between patients with NF1 and suspected NF1, and patients with suspected NF1 and controls were observed (both $P > 0.05$). In the third choroidal layer, the flow area was significantly lower in patients with NF1 compared to those with suspected NF1 (49.7 ± 6.7 vs 54.3 ± 5.7%; $P = 0.013$); while no differences between patients with NF1 and controls, and patients suspected NF1 and controls were observed (both $P > 0.05$).

SS-OCTA images revealed the presence of 'corkscrew' vessels in 9 patients with NF1 (29.0%), and in none of the patients with suspected NF1 and control subjects ($P < 0.001$). These abnormal vessels were located in the posterior pole along the vascular arcades. No patients with NF1 presented more than one 'corkscrew' vessel for each eye. Moreover, these alterations were also found by using SS-OCT-en face, appearing as tortuous vessels parallel to the normal arteriosus profile.

**Table 3. Vascular flow area of different retinal and choroid layers in NF1 patients, suspected NF1 patients and controls.**

|  | NF1 (n = 31) | Suspected NF1 (n = 25) | Controls (n = 34) | $P^{*}$ |
|---|---|---|---|---|
| SCP (%) | 42.6 ± 2.2 | 43.1 ± 2.5 | 41.0 ± 2.0 | 0.002 |
| DCP (%) | 44.3 ± 3.0 | 45.7 ± 2.8 | 44.6 ± 2.9 | 0.185 |
| CC (%) | 53.5 ± 3.0 | 54.5 ± 3.3 | 53.7 ± 1.3 | 0.352 |
| Choroid 1 (%) | 51.1 ± 3.5 | 51.1 ± 4.1 | 52.8 ± 3.1 | 0.113 |
| Choroid 2 (%) | 45.4 ± 4.8 | 47.1 ± 5.7 | 49.0 ± 4.0 | 0.015 |
| Choroid 3 (%) | 49.7 ± 6.7 | 54.3 ± 5.7 | 51.6 ± 5.2 | 0.018 |

NF1: Neurofibromatosis type 1; SCP: Superficial capillary plexus; DCP: Deep capillary plexus; CC: Choriocapillaris.

[*] One-way ANOVA.

## Discussion

To the best of our knowledge, this is the first prospective, large scale study that aimed to simultaneously analyze both choroidal and retinal thickness variations by SS-OCT, the presence of choroidal nodules and nonetheless, the alterations in the vascular flow area by SS-OCTA in patients with NF1.

Choroidal nodules consist of proliferating Schwann cells arranged in concentric rings and show histological similarities to cutaneous neurofibromata and Lisch nodules [15, 16]. In the present study, choroidal nodules were identified in 77.4% of patients with NF1 and 8.0% of patients with suspected NF1. These results are in line with those of previous studies, which reported a frequency of choroidal nodules in pediatric patients with NF1 ranging from 69 to 71% (3,8). The slightly higher frequency of choroidal nodules in our study may be caused by the older sample size examined. Moreover, choroidal nodules were more specific for NF1 compared to Lisch nodules. This confirms the utility of choroidal nodules as a new diagnostic marker of NF1.

In the present study, we found a significant reduction of macular RNFL thickness and retinal thickness in patients with NF1 compared to those with suspected NF1 and control subjects. A reduced macular RNFL thickness was previously reported in patients with NF1 and optic pathway gliomas [17, 18]. Thus, the measurement of macular RNFL thickness has been proposed as a useful tool to screen NF1 patients for the presence of gliomas [15, 17]. However, only 3 patients in our cohort had a optic pathway glioma. In this regard, as previously reported in literature, RNFL thinning has been found also in adult patients with NF1 without the presence optic nerve glioma [16]. Hence, our results are in line with these findings suggest that macular RNFL thickness may be reduced in patients with NF1 even in the absence of this complication. Further studies in the near future should better clarify if the measurement of RNFL and GCL thicknesses may be helpful in the management of patients with NF1 without glioma.

Although a significant decrease of the choroidal vascular flow area was found in NF1 patients with OCTA, we did not identify any significant difference in choroidal thickness among patients with NF1, suspected NF1 and controls. This result is in contrast with those of Abdolrahimzadeh et al., who reported significant choroidal thinning in adults patients with NF1 [16]. The difference may be due to the younger age of patients in our study. Therefore, that the reduced choroidal circulation measured with OCTA might be an earlier sign of the choroidal involvement in NF1, while choroidal thinning may occur later in life.

The association between NF1 and RVAs has been widely demonstrated. Muci-Mendoza et al. described RVAs as small tortuous venules with a "corkscrew" appearance and revealed that their prevalence was 37.5% of the patients with NF1 by performing direct ophthalmoscopy or FA [11] Abdolrahimzadeh et al. found in 35% of the patients with NF1 the presence of RVAs overlying the choroidal abnormalities detected by the contemporary use of NIR and EDI-OCT [12]. In agreement with these results, by using SS-OCTA we were able to identify RVAs in 29% of patients with NF1 and in none of the patients with suspect NF1 and control subjects.

The introduction of OCTA in the clinical practice has allowed us to demonstrate the laminar organization of the capillary plexus, in which the DCP is composed by polygonal units draining into the superficial venules of the SCP [17]. Parrozzani et al. detected RVAs in only 6.1% of the patients with NF1 by OCTA; in this regard, they found that the presence of RVAs was not related to patients' visual outcome. In addition, they found that all the RVAs were situated in the SCP, while in 75% of the cases they were associated with a congested capillary network in the DCP. The authors postulated that the primary damage would reside in DCP

changes, bringing to the formation of small crowded arterial vessels draining into the SCP [18].

Our study was the first one to investigate the vascular flow area measured by OCTA in patients with NF1. Interestingly, NF1 patients showed a higher vascular flow area of the SCP as compared with healthy subjects and patients suspected NF1. The presence of RVAs in the SCP could explain this finding. In addition, patients with NF1 showed a significant decrease of the vascular flow area both in the second and third choroidal layers. This is in line with previous studies demonstrating an altered choroidal blood perfusion by ICGA in patients with NF1, with hypofluorescent areas in the CC found in the early phases [9]. Thus, the altered vascular flow area at the different choroidal layers in patients with NF1 may be due to a pathological redistribution of the vascular flow caused by the presence of choroidal lesions.

In conclusion, this present study showed the significant alterations in retinal and macular RNFL thicknesses, a high prevalence of choroidal nodules, and, nonetheless, the altered retinal and choroidal vascular flow area measured by OCTA in NF1 patients. Further larger scale studies adopting this novel device should better clarify the complex pathological mechanisms involved in these vascular alterations.

## Author Contributions

**Conceptualization:** Aldo Vagge, Paola Camicione, Giulia Agosto, Giuseppe Giannaccare, Massimo Nicolò, Carlo Enrico Traverso.

**Data curation:** Aldo Vagge, Paolo Corazza, Lorenzo Ferro Desideri, Paola Camicione, Giulia Agosto, Calevo Maria Grazia, Carlo Enrico Traverso.

**Formal analysis:** Aldo Vagge, Paolo Corazza, Calevo Maria Grazia, Adriano Carnevali, Carlo Enrico Traverso.

**Funding acquisition:** Aldo Vagge.

**Investigation:** Paolo Corazza, Lorenzo Ferro Desideri, Roberta Vagge, Massimo Nicolò.

**Methodology:** Paolo Corazza, Lorenzo Ferro Desideri, Roberta Vagge, Adriano Carnevali, Massimo Nicolò.

**Resources:** Carlo Enrico Traverso.

**Supervision:** Aldo Vagge, Massimo Nicolò, Carlo Enrico Traverso.

**Validation:** Aldo Vagge, Carlo Enrico Traverso.

**Writing – original draft:** Lorenzo Ferro Desideri.

**Writing – review & editing:** Aldo Vagge, Adriano Carnevali, Giuseppe Giannaccare, Massimo Nicolò, Carlo Enrico Traverso.

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
