## [Decision Letter · Decision Letter 0]

25 Feb 2021

PONE-D-21-01698

Ocular biometric parameters changes and choroidal vascular abnormalities in patients with neurofibromatosis type 1 evaluated by OCT-A

PLOS ONE

Dear Dr. Ferro Desideri,

Thank you for submitting your manuscript to PLOS ONE. After careful consideration, we feel that it has merit but does not fully meet PLOS ONE’s publication criteria as it currently stands. Therefore, we invite you to submit a revised version of the manuscript that addresses the points raised during the review process.

Please correct referencing and formatting errors. Please provide brain MRI images or the results of brain MRI for NF1 patients. Focus on your specific results and don't conclude that this can be a diagnostic method at this stage.

We look forward to receiving your revised manuscript.

Kind regards,

Alfred S Lewin, Ph.D.

Academic Editor

PLOS ONE

2. Thank you for including your ethics statement:  "Informed consent was obtained from all participants and/or their familiesandthis study was done in accordance with the Declaration of Helsinki and was approved by the Institutional Ethics Committee".   

a.) Please amend your current ethics statement to include the full name of the ethics committee/institutional review board(s) that approved your specific study.

b.) Please provide additional details regarding participant consent. In the ethics statement in the Methods and online submission information, please ensure that you have specified what type you obtained (for instance, written or verbal, and if verbal, how it was documented and witnessed). If your study included minors, state whether you obtained consent from parents or guardians. If the need for consent was waived by the ethics committee, please include this information.

<h1>** **</h1>

Reviewers' comments:

Reviewer's Responses to Questions

**Comments to the Author**

1. Is the manuscript technically sound, and do the data support the conclusions?

Reviewer #1: Yes

Reviewer #2: Yes

2. Has the statistical analysis been performed appropriately and rigorously? 

Reviewer #1: Yes

Reviewer #2: Yes

3. Have the authors made all data underlying the findings in their manuscript fully available?

Reviewer #1: Yes

Reviewer #2: Yes

4. Is the manuscript presented in an intelligible fashion and written in standard English?

Reviewer #1: Yes

Reviewer #2: Yes

5. Review Comments to the Author

Reviewer #1: This is an interesting paper in an area of active research (biomarkers for NF1) that has rapidly progressed with multimodal imaging. The use of swept source and OCTA by the authors provides further knowledge on the topic.

Some revision would greatly enhance the manuscript as follows:

Abstract

Specify “macular” RNFL, if this is the case. This needs to be specified in the methods section and throughout.

In the conclusions avoid general comments (first sentence) and focus on specific results such as the major novel finding ie choroidal and retinal vascular flow area. The use of the term “diagnostic tool” may be appropriate once further studies are carried out, rather the findings of the authors provide further knowledge on the vascular alterations of the choroid using advance imaging, and thinning of the RNFL even in the absence of optic nerve gliomas

In the results section please check language: the word “compared” is inserted twice

Introduction

Although the diagnosis of NF1 is clinical, some centers use genetic testing for assessment, please mention this.

Please correct to: …contemporary presence of “the” following criteria…

Please correct to: “The presence of choroidal nodules, detected as bright patchy areas by NIR, has previously been reported in 70% of pediatric patients with NF1.”

Please correct to: Furthermore, in patients with NF1, the association between choroidal nodules and retinal vascular abnormalities (RVAs) has been reported.

Please correct: OCTA has “enabled” us

Methods

Please state if IOP was measured and please state if patients with glaucoma or borderline IOP were excluded, as this can influence RNFL and GCL- IPL thickness values.

Please correct: Remove the word “Furthermore”…… 3 different levels (l1, L2, and L3…)

Results

Please evaluate RNFL and GCL excluding patients with glioma in order to confirm significance results. If, following statistical analysis, the results change please comment accordingly in discussion.

Discussion

The authors state: “A reduced RNFL thickness was previously reported in patients with NF1 and optic pathway gliomas (17,18). Thus, the measurement of RNFL thickness has been proposed as a useful tool to screen NF1 patients for the presence of gliomas (17). However, only 3 patients in our cohort had a optic pathway glioma. Therefore, RNFL thickness may be reduced in patients with NF1 even in the absence of this complication.” There are publications on RNFL thinning in patients with optic nerve glioma (Chang L, et al. Optical coherence tomography in the evaluation of neurofibromatosis type-1 subjects with optic pathway gliomas. J AAPOS 2010;14:511-517.; Topcu-Yilmaz P, et al. Investigation of retinal nerve fiber layer thickness in patients with neurofibromatosis-1. Jpn J Ophthalmol 2014;88:172-176.)

but also RNFL thinning in adult patients WITHOUT optic nerve glioma (Abdolrahimzadeh S, et al. Spectral domain optical coherence tomography evidence of retinal nerve fibre layer and ganglion cell loss in adult patients with neurofibromatosis type 1. Retina 2016; 36:75-81). Furthermore, some authors have studied the peripapillary RNFL and some also the macular RNFL.

This section would benefit with some discussion, please mention the relevant references above when discussing the results on RNFL thickness.

Citations and references

Please correct citations in text: in some sections the citation in the text is the author name and year of publication, in other sections there are numbers. Furthermore, references are not identified by numbers in the reference section. Kindly correct citations and references according to journal instructions.

Reference 16 is missing in the reference section when the authors refer to the previous study on choroidal thickness, Br J Ophthalmol 2015;99:789-793.

References 17, and 18 are missing in the reference section when the authors refer to RNFL thinning, Chang L et al and Topcu-Yilmaz P et al.

Please carefully check citations and reference list for accuracy throughout.

Reviewer #2: Very interesting study

one major remark: No data on MRI of the brain was given, if not all NF1 patients and NF1 suspected patients have undergone an MRI, no conclusions related to the RNFL can be drawn

6. PLOS authors have the option to publish the peer review history of their article (what does this mean?). If published, this will include your full peer review and any attached files.

Reviewer #1: No

Reviewer #2: No

---

## [Author Response · Author response to Decision Letter 0]

10 Apr 2021

Reviewer #1: This is an interesting paper in an area of active research (biomarkers for NF1) that has rapidly progressed with multimodal imaging. The use of swept source and OCTA by the authors provides further knowledge on the topic.

Some revision would greatly enhance the manuscript as follows:

Abstract

Specify “macular” RNFL, if this is the case. This needs to be specified in the methods section and throughout.

We thank the reviewer for this comment; we specified it was the RFNL thickness measured in the macular region.

In the conclusions avoid general comments (first sentence) and focus on specific results such as the major novel finding ie choroidal and retinal vascular flow area. The use of the term “diagnostic tool” may be appropriate once further studies are carried out, rather the findings of the authors provide further knowledge on the vascular alterations of the choroid using advance imaging, and thinning of the RNFL even in the absence of optic nerve gliomas

We thank the reviewer for this insight. We avoided general comments as suggested, focusing on the possible importance of our findings and we agree with the fact that OCT-A ca not be considered a diagnostic tool yet considering the relatively small evidence in this regard to date.

In the results section please check language: the word “compared” is inserted twice

We corrected it as suggested.

Introduction

Although the diagnosis of NF1 is clinical, some centers use genetic testing for assessment, please mention this.

We specified it as suggested by the reviewer

Please correct to: …contemporary presence of “the” following criteria…

We modified it as suggested by the reviewer

Please correct to: “The presence of choroidal nodules, detected as bright patchy areas by NIR, has previously been reported in 70% of pediatric patients with NF1.”

We modified it as suggested

Please correct to: Furthermore, in patients with NF1, the association between choroidal nodules and retinal vascular abnormalities (RVAs) has been reported.

We modified it as suggested

Please correct: OCTA has “enabled” us

We modified it as suggested by the reviewer.

Methods

Please state if IOP was measured and please state if patients with glaucoma or borderline IOP were excluded, as this can influence RNFL and GCL- IPL thickness values.

Please correct: Remove the word “Furthermore”…… 3 different levels (l1, L2, and L3…)

We added the corrections suggested

Results

Please evaluate RNFL and GCL excluding patients with glioma in order to confirm significance results. If, following statistical analysis, the results change please comment accordingly in discussion.

We thank the reviewer for this comment. As correctly stated by reviewer cases of gliomas were present in our cohort; however, considering the post chiasmatic/optic radiations position of the finding, we did not rule out these patients from the statistical analysis. In this regard, no previous evidence has been reported in literature about a possible decrease in vascular density by OCTA in the choroid of patients with postchiasmatic gliomas. Moreover, we deem that n=3/31 patients does not represent a significant number of patients interfering with the significance of the findings in our study.

Discussion

The authors state: “A reduced RNFL thickness was previously reported in patients with NF1 and optic pathway gliomas (17,18). Thus, the measurement of RNFL thickness has been proposed as a useful tool to screen NF1 patients for the presence of gliomas (17). However, only 3 patients in our cohort had a optic pathway glioma. Therefore, RNFL thickness may be reduced in patients with NF1 even in the absence of this complication.” There are publications on RNFL thinning in patients with optic nerve glioma (Chang L, et al. Optical coherence tomography in the evaluation of neurofibromatosis type-1 subjects with optic pathway gliomas. J AAPOS 2010; 14:511-517.; Topcu-Yilmaz P, et al. Investigation of retinal nerve fiber layer thickness in patients with neurofibromatosis-1. Jpn J Ophthalmol 2014; 88:172-176.)

but also RNFL thinning in adult patients WITHOUT optic nerve glioma (Abdolrahimzadeh S, et al. Spectral domain optical coherence tomography evidence of retinal nerve fibre layer and ganglion cell loss in adult patients with neurofibromatosis type 1. Retina 2016; 36:75-81). Furthermore, some authors have studied the peripapillary RNFL and some also the macular RNFL.

This section would benefit with some discussion, please mention the relevant references above when discussing the results on RNFL thickness.

We thank the reviewer for this comment. We added the suggested studies dealing with the study of RNFL thickness in NF1 patients without glioma, discussing its potential role for further studies.

Citations and references

Please correct citations in text: in some sections the citation in the text is the author name and year of publication, in other sections there are numbers. Furthermore, references are not identified by numbers in the reference section. Kindly correct citations and references according to journal instructions.

We modified the references style as suggested by the reviewer

Reference 16 is missing in the reference section when the authors refer to the previous study on choroidal thickness, Br J Ophthalmol 2015;99:789-793.

References 17, and 18 are missing in the reference section when the authors refer to RNFL thinning, Chang L et al and Topcu-Yilmaz P et al.

Please carefully check citations and reference list for accuracy throughout.

All the references have been carefully checked and reordered

Reviewer #2: Very interesting study

one major remark: No data on MRI of the brain was given, if not all NF1 patients and NF1 suspected patients have undergone an MRI, no conclusions related to the RNFL can be drawn

We thank the reviewer for the comment. We deem that our study has an ophthalmological perspective and the evaluation of MRI data, beside the individuation of 3 postchiasmatic gliomas in 3 out 31 patients with confirmed diagnosis of NF1, falls outside the topic of this study. Moreover, MRI findings are not very unlikely to interfere with the alterations reported in choroidal vasculature and RNFL thinning measure by OCT. 

We better discussed our findings dealing with RNFL thinning in the discussion paragraph.

---

## [Editor Report · Decision Letter 1]

20 Apr 2021

Ocular biometric parameters changes and choroidal vascular abnormalities in patients with neurofibromatosis type 1 evaluated by OCT-A

PONE-D-21-01698R1

Dear Dr. Ferro Desideri,

We’re pleased to inform you that your manuscript has been judged scientifically suitable for publication and will be formally accepted for publication once it meets all outstanding technical requirements.

Kind regards,

Alfred S Lewin, Ph.D.

Section Editor

PLOS ONE
---

## [Editor Report · Acceptance letter]

26 Apr 2021

PONE-D-21-01698R1 

Ocular biometric parameters changes and choroidal vascular abnormalities in patients with neurofibromatosis type 1 evaluated by OCT-A 

Dear Dr. Ferro Desideri:

I'm pleased to inform you that your manuscript has been deemed suitable for publication in PLOS ONE. Congratulations! Your manuscript is now with our production department. 

Kind regards, 

on behalf of

Dr. Alfred S Lewin 

Section Editor

PLOS ONE